

# Correlations and linewidth of the atomic beam continuous superradiant laser

Bruno Laburthe-Tolra*, Ziyad Amodjee,
Benjamin Pasquiou 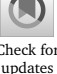 and Martin Robert-de-Saint-Vincent 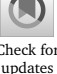

CNRS, UMR 7538, LPL, F-93430, Villetaneuse, France
Laboratoire de Physique des Lasers, Université Sorbonne Paris Nord,
F-93430, Villetaneuse, France

* bruno.laburthe-tolra@univ-paris13.fr

## Abstract

We propose a minimalistic model to account for the main properties of a continuous superradiant laser, in which a beam of atoms crosses the mode of a high-finesse Fabry-Perot cavity, and collectively emits light into the cavity mode. We focus on the case of weak single atom - cavity cooperativity, and highlight the relevant regime where decoherence due to the finite transit time dominates over spontaneous emission. We propose an original approach where the dynamics of atoms entering and leaving the cavity is described by a Hamiltonian process. This allows deriving the main dynamical equations for the superradiant laser, without the need for a stochastic approach. We derive analytical conditions for a sustained emission and show that the ultimate linewidth is set by the fundamental quantum fluctuations of the collective atomic dipole. We calculate steady-state values of the two-body correlators and show that the continuous superradiant regime is tied to the growth of atom-atom correlations, although these correlations only have a small impact on the laser linewidth.

# 1 Introduction

The prospect for a continuous superradiant laser, in which an ensemble of atoms continuously and cooperatively emits light in a cavity mode [1], has attracted attention because of the fundamental interest in cavity quantum electrodynamics and open quantum many-body systems [2], and because of potential metrological applications [3,4]. Indeed, these lasers operate deep in the bad-cavity limit, such that the frequency of the laser is only very weakly sensitive to mirror vibrations, and is instead mainly set by the natural frequency of the atomic transition. In addition, in the continuous regime, the laser's linewidth could even reach values below the natural linewidth of the transition - a feature that strongly contrasts with the case of pulsed superradiance, for which the linewidth is typically increased [5,6]. For these reasons, it is believed that such systems could become a new architecture for an atomic clock, an architecture where a self-referenced ultra-stable laser is the clock itself [7,8]. First experimental realisations have been achieved [7,9–12], although up to now none of these have reached the continuous regime (see however [13] for a beam of atoms with pre-defined correlations).

To reach the continuous regime, two approaches are typically discussed in the literature: (i) ultra-cold atoms are trapped in a cavity and continuously repumped to an excited electronic state [4]; (ii) a flux of atoms in an excited state crosses the cavity mode [14]. Here, we focus on the second architecture. We introduce an *ab-initio* theory starting from a purely Hamiltonian description of atom in- and out-coupling, and photon out-coupling. From this we derive all the main properties of the superradiant laser, without introducing fluctuating variables or jump operators to derive a master equation. This approach allows to derive analytical equations with explicit assumptions, that provide the criteria for continuous superradiance, the dynamics of the coupled atom - cavity system, and the power of the emitted radiation. Furthermore, we calculate the second moments of the atomic operators. We show that continuous superradiance is obtained concurrently with the build-up of atom-atom correlations. We use this result to derive an analytical expression of the laser linewidth when operating deep in the bad-cavity limit.

# 2 Hamiltonian description of a pulsed superradiant laser

Most descriptions of superradiant lasers rely on Langevin models for the atomic and cavity field operators. While quantum Langevin equations are used in several studies of continuously repumped superradiant lasers (e.g. [15, 16]), recent works on the atomic beam architecture use stochastic classical Langevin equations, based on complex-number descriptions of the operators [14, 17, 18]. Stochastic terms are associated with the description of cavity leakage, spontaneous emission and atomic dephasing. Stochastic initial values of the atomic operators are also required to model two-body atomic correlations [14, 17, 19]. The physical properties are then obtained through Monte Carlo calculations. By contrast, we base our work on a Hamiltonian quantum description, without stochastic terms. Our aims are to present a self-contained theoretical description, and to provide new insights on the laser threshold, power, correlation properties, and linewidth.

## 2.1 Derivation of the effective Hamiltonian

At first we assume that the atoms can only emit light in the cavity mode, which corresponds to neglecting spontaneous emission towards all the other electromagnetic modes. We also assume the cavity mode to be resonant with the atomic transition. The interaction of $N$ atoms identically coupled to the cavity mode is described by the following Hamiltonian, written in the interaction picture:

$$H_0 = g \sum_{i=1}^{N} \left( s_i^- b^+ + s_i^+ b \right), \tag{1}$$

where $g$ is the single-atom light-cavity interaction parameter and $b$ the destruction operator for a photon of the cavity mode. The atomic degrees of freedom are described by spin operators $s_i^{+,-,z}$ associated with the two electronic levels of each atom $i$, $|g_i\rangle$ and $|e_i\rangle$: we have $s_i^+ = |e_i\rangle\langle g_i|$, $s_i^- = |g_i\rangle\langle e_i|$, $s_i^z = \frac{1}{2}(|e_i\rangle\langle e_i| - |g_i\rangle\langle g_i|)$, and $s_i^x = \frac{1}{2}\left(s_i^+ + s_i^-\right)$, $s_i^y = \frac{1}{2i}\left(s_i^+ - s_i^-\right)$.

To describe cavity losses, we consider that the cavity mode is coherently coupled to the continuum of states made of the many longitudinal modes outside the cavity that share the transverse mode set by the cavity. These modes are described by the destruction operator $a_k$, and their detuning with respect to the cavity mode $\omega_k$. The coherent coupling $\Omega$ of the cavity mode to all these modes is assumed to be independent of $k$. This leads to the following Hamiltonian:

$$H = H_0 + \Omega \sum_k \left( b a_k^+ \exp(i\omega_k t) + b^+ a_k \exp(-i\omega_k t) \right). \tag{2}$$

We have (setting $\hbar = 1$):

$$i\frac{d\langle a_k\rangle}{dt} = \langle [a_k, H] \rangle = \Omega \langle b \rangle \exp(i\omega_k t), \tag{3}$$

$$\langle a_k \rangle = a_k^0 + \frac{1}{i} \int_0^t d\tau \, \Omega \langle b(\tau) \rangle \exp(i\omega_k \tau), \tag{4}$$

with $a_k^0 = \langle a_k \rangle (t = 0)$. Unless stated otherwise, brackets correspond to the quantum expectation value. Making a mean-field approximation on the output cavity field, we find:

$$H \approx H_0 + \Omega \sum_k \left( b^+ \frac{1}{i} \int_0^t d\tau \, \Omega \langle b \rangle (\tau) \exp(i\omega_k(\tau - t)) + h.c \right)$$
$$+ \Omega \sum_k \left( b^+ a_k^0 \exp(-i\omega_k t) + h.c. \right). \tag{5}$$

Here, as $\langle b \rangle (\tau)$ varies slowly, we make a standard approximation $\langle b \rangle (\tau) \to \langle b \rangle (t)$:

$$H \approx H_0 + \frac{\Omega^2}{i} \sum_k \left( b^+ \langle b \rangle (t) \int_0^t d\tau \exp(i\omega_k(\tau - t)) - h.c \right) + \Omega \sum_k \left( b^+ a_k^0 \exp(-i\omega_k t) + h.c. \right).$$

We have:

$$\int_0^t d\tau \exp(i\omega_k(\tau - t)) = \frac{-i}{\omega_k} (1 - \cos(\omega_k t)) + \frac{1}{\omega_k} (\sin(\omega_k t)).$$

The first term is odd in $\omega_k$ so that the summation over $k$ is zero, and the associated frequency shift is therefore neglected. On the other hand, $\sin(\omega_k t)/\omega_k \to \pi\delta(\omega_k)$ at long times. Therefore:

$$H \approx H_0 + \frac{\Omega^2}{i} \pi\mathcal{N} \left( b^+ \langle b \rangle - b \langle b^+ \rangle \right) + f(t), \tag{6}$$

where we have defined $f(t) = \Omega \sum_k \left( b^+ a_k^0 \exp(-i\omega_k t) + h.c. \right)$ and $\mathcal{N}$ is the density of states of the external coupled modes. Eq. (6) is the mean-field version of the stochastic Langevin equation and $f(t)$ formally corresponds to its stochastic term. Here, since the evolution is purely Hamiltonian, this term simply arises from the initial condition on the output field [20]. Here, and in Section 3 where we consider a deterministic reloading of the cavity, we consider the case where the output field is initially empty, leading to $a_k^0 = 0$ and therefore $f(t) = 0$. We do not consider other outcoupling or dissipative mechanism, that would lead to the need for additional stochastic terms. Finally, we have:

$$H = H_0 + \frac{\Omega^2}{i} \pi\mathcal{N} \left( b^+ \langle b \rangle - b \langle b^+ \rangle \right). \tag{7}$$

We define the cavity leakage rate

$$\kappa = 2\pi\Omega^2 \mathcal{N}. \tag{8}$$

## 2.2 Equations for the mean values of the operators

We will here derive the equation of evolution of expectation values for all field and atomic operators within a mean-field approximation. We have:

$$i\frac{d\langle b \rangle}{dt} = \langle [b, H] \rangle = g \sum_{i=1}^N \langle s_i^- \rangle + \frac{\kappa}{2i} \langle b \rangle. \tag{9}$$

Likewise,

$$i\frac{d\langle s_i^- \rangle}{dt} = g \langle [s_i^-, s_i^+] b \rangle = -2g \langle s_i^z b \rangle. \tag{10}$$

Here we neglect correlations between the atomic degrees of freedom and the cavity field, by writing: $-2g \langle s_i^z b \rangle \approx -2g \langle s_i^z \rangle \langle b \rangle$. The last part of this paper (Section 5) goes beyond this mean-field approximation. Defining $S^{+,-,z} = \sum_i s_i^{+,-,z}$, we find:

$$i\frac{d\langle S^- \rangle}{dt} \approx -2g \langle S^z \rangle \langle b \rangle. \tag{11}$$

Likewise, we find

$$i\frac{d\langle S^z \rangle}{dt} \approx -g \langle S^- \rangle \langle b^+ \rangle + g \langle S^+ \rangle \langle b \rangle. \tag{12}$$

We perform a last approximation, which is to assume an adiabatic elimination of the cavity field:

$$i\frac{d\langle b\rangle}{dt} = 0\,,$$

from which we deduce, using Eq. (9),

$$\langle b\rangle = -2i\frac{g}{\kappa}\langle S^-\rangle\,. \tag{13}$$

As shown by Eq. (9), the cavity field $b$ relaxes towards its steady state in a timescale $1/\kappa$. The adiabatic elimination is therefore valid if $1/\kappa$ is much smaller than the timescale of evolution of atomic variables.

From Eqs. (9), (11), and (12), we find the following set of equations for the collective atomic degrees of freedom:

$$\begin{aligned}
\frac{d\langle S^z\rangle}{dt} &= -4\frac{g^2}{\kappa}\langle S^-\rangle\langle S^+\rangle\,,\\
\frac{d\langle S^-\rangle}{dt} &= 4\frac{g^2}{\kappa}\langle S^z\rangle\langle S^-\rangle\,,\\
\frac{d\langle S^+\rangle}{dt} &= 4\frac{g^2}{\kappa}\langle S^z\rangle\langle S^+\rangle\,.
\end{aligned} \tag{14}$$

Eqs. (14) possess analytical solutions, and remarkably depend only on the parameter $g^2/\kappa$. We point out that $\langle S^x\rangle^2 + \langle S^y\rangle^2 + \langle S^z\rangle^2$ is a conserved quantity, therefore superradiance can generically be described on a simple Bloch sphere [10]. As is expected for a mean-field equation, this set of equations is in a metastable configuration if initially all atoms are initialised in the excited state, with $\langle S^z\rangle = N/2$ and $\langle S^+\rangle = \langle S^-\rangle = 0$. However, it does describe a superradiant burst if initially a non-vanishing dipole is assumed, $e.g.$ $\langle S^+\rangle \neq 0$, see Fig. 1. Qualitatively, when there are $N$ atoms in the cavity, $\langle S^z\rangle$ is of order $N$, such that the dipole $\langle S^+\rangle$ decays in a timescale $(Ng^2/\kappa)^{-1}$. The factor $N$ comes from the collective nature of super-radiant emission [5]. The adiabatic elimination introduced above thus requires $\kappa \gg Ng^2/\kappa$, $i.e.$, $\kappa \gg \sqrt{N}g$.

## 2.3 Spectrum of the pulsed superradiant laser

After solving the atoms' dynamics using Eqs. (14), one can simply calculate the emitted spectrum using Eq. (4) ($i.e.$ the amplitude of the field as a function of $\omega_k$). Here, we start with an almost fully inverted situation with $\langle S^z\rangle \approx N/2$, and a microscopic but non-vanishing dipole in order to avoid the metastable situation that arises at the mean-field level when $\langle S^-\rangle = 0$. Fig. 1 shows that $\langle S^-\rangle$ peaks after a finite time, which corresponds to a superradiant burst. At the same time atoms decay from the upper to the lower state, so that $\langle S^z\rangle$ asymptotically reaches $-N/2$. At short times, the emitted spectrum has a width that is found to numerically scale as $1/t$. After a time $t_N \approx \kappa/Ng^2$, all atoms have decayed to the ground state, the laser ceases to emit, and the width of the radiation stops reducing. Therefore, the width of a pulsed superradiant laser is inherently set by $Ng^2/\kappa$ [5]. In other words, the laser spectrum is at best Fourier limited by the pulse envelope. For metrological applications, it can therefore be useful to reach a sustained or CW regime, in order to further reduce the linewidth.

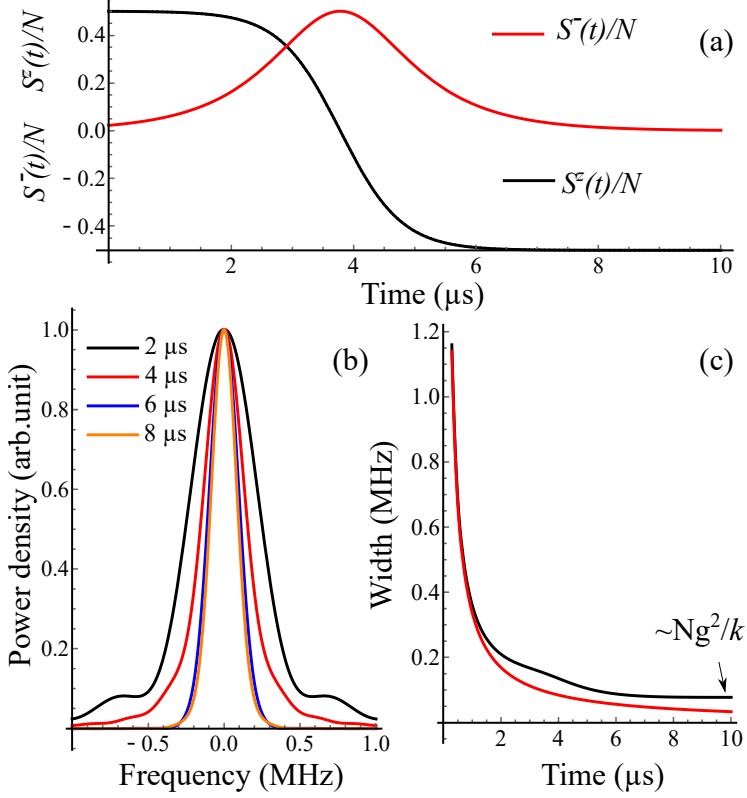

Figure 1: Pulsed superradiance. (a) evolution of the collective operators $\langle S^- \rangle$ and $\langle S^z \rangle$ ; (b) spectrum at 2, 4, 6, and 8 microseconds; (c) black: narrowing of the spectrum as a function of time. A $1/t$ behaviour, exemplified in red, is seen at very short times (although corresponding to very little light emission). A further reduction is obtained during the superradiant pulse. As the light intensity varies in time, the $1/t$ scaling of the linewidth is then only qualitative. An asymptotic value of $Ng^2/\kappa$ is seen at long times, corresponding to the inverse of the duration of the burst. These results are obtained for $g/2\pi = 4 \times 10^3$ Hz, $\kappa/2\pi = 2 \times 10^5$ Hz, and $N = 10^3$ atoms.

# 3 Hamiltonian description of a continuous superradiant laser

## 3.1 Atom in- and out-coupling

We now add to our description the possibility to load and unload atoms in the cavity mode, so that continuous operation becomes possible. The loading and unloading of atoms in the cavity can either be a stochastic process (as will inevitably happen in a beam experiment [13]) or a deterministic process (which may happen if atoms are loaded *e.g.* by using a moving trap [21]). Here we will assume that the loading and unloading is a deterministic process. The interest of this approach is that the superradiant laser can still be treated using a fully Hamiltonian description, without dissipation. Our approach will be to consider that the coupled atomic+cavity+light system is at all times in a pure state, which describes a (potentially infinite) number of atoms. The averages $\langle . \rangle$ below are expectation values over this pure quantum mechanical state. At any given time, the number of atoms inside the cavity mode is finite. We will consider *time-dependent* operators, that describe how an individual atom is coupled to the cavity mode only within a specific time-span [22, 23]. As we shall see, this will allow deriving the superradiant laser equations without resorting to the usual master equation in the Born-Markov approximation, with Liouvillian operators for stochastic loading and unloading.

The atom-cavity interaction is now described as

$$H_0(t) = g \sum_{j=1}^{\infty} \left( \eta_j(t) s_j^- b^+ + \eta_j(t) s_j^+ b \right), \tag{15}$$

where $\eta_j(t)$ is a real function that is 0 when atom $j$ is outside of the cavity and 1 when the atom is inside. At a given time $t_j$, the atom $j$ enters the cavity, and $\eta_j$ quickly and linearly raises from 0 to 1 in a time $\tau_0$. At the time $t_j + N/\Gamma$, the atom $j$ leaves the cavity, and $\eta_j$ linearly goes to 0 in a time $\tau_0$. Thus, within the time interval $\left[ t_j, t_j + \tau_0 \right]$, we assume that atom $j$ enters the cavity, and atom $j-N$ leaves the cavity. Here, $\Gamma$ is the loading rate of the cavity, $\Gamma_R = \Gamma/N$ is the refreshing rate of the cavity, *i.e.*, the inverse of the transit time, with a steady-state number $N$ of atoms inside the cavity. We do not expect that the actual mathematical form chosen for $\eta_j(t)$ qualitatively impacts the result of our analysis.

In practice, we propose to re-define the spin raising and lowering operators with the following rule:

$$s_j^{z,+,-}(t) = \eta_j(t) \times s_j^{z,+,-}. \tag{16}$$

The commutation rules are modified accordingly, e.g.

$$\left[ s_j^-(t), s_j^+(t) \right] = -2\eta_j(t) s_j^z(t) = -2\eta_j(t)^2 s_j^z. \tag{17}$$

We now calculate the dynamical equation of the atomic operators inside the cavity $S^\epsilon(t) = \sum_{j=0}^{\infty} \eta_j(t) s_j^\epsilon$ ($\epsilon = (+,-,z)$). Eq. (9) on $\langle b \rangle$ is unchanged; Eq. (4), that allows calculating the emitted spectrum of light, is also unchanged. However, the equations on the atomic variables are modified because the spin operators are now time-dependent. Using the general equation $i\frac{d\langle A \rangle}{dt} = \langle [A,H] \rangle + i\left\langle \frac{\partial A}{\partial t} \right\rangle$, we have

$$i\frac{d\left\langle S^-(t) \right\rangle}{dt} = -2g \left\langle \sum_{j=0}^{\infty} \eta_j(t) s_j^z(t) b \right\rangle + i \sum_{j=0}^{\infty} \frac{d\eta_j(t)}{dt} \left\langle s_j^- \right\rangle. \tag{18}$$

Since the loading and unloading of individual atoms is operated at a rate $\Gamma$, we study the evolution of the atomic variables for a duration $1/\Gamma$. Outside of the time intervals $\left[ t_j, t_j + \tau_0 \right]$ Eq. (18) is identical to Eq. (11), because then all values of $\eta_j(t)$ are either 0 or 1. Therefore, one only needs to consider the modifications associated with the time evolution driven by Eq. (18) when $t$ is in the interval $\left[ t_j, t_j + \tau_0 \right]$. If $\tau_0$ is small enough, one can assume that during this time interval, comparatively to the case without atom refreshing, $\langle S^- \rangle$ is simply modified by $\delta S^-_{|load,j}$ given by

$$i\delta S^-_{|load,j} = -2g \int_{t_j}^{t_j+\tau_0} dt' \left\langle b s_j^z \right\rangle \eta_j^2(t') - 2g \int_{t_j}^{t_j+\tau_0} dt' \left\langle b s_{j-N}^z \right\rangle \eta_{j-N}^2(t') + i\left( \left\langle s_j^- \right\rangle - \left\langle s_{j-N}^- \right\rangle \right). \tag{19}$$

If $\tau_0 \to 0$ the first two terms of this equation vanish, as their integrand is finite. We will also assume that the atom entering the cavity has been prepared in the excited state, such that $\left\langle s_j^- \right\rangle = 0$. Furthermore, we also make the following key assumption: $\left\langle s_{j-N}^- \right\rangle = \langle S^- \rangle/N$. This assumption requires either that each atom reaches a steady state shortly after entering the cavity, or that the many atoms entering the cavity (in practice at random times and random velocities) follow different trajectories, such that, at the exit, the statistical mean for such random realisations is identical to the average inside the cavity (ergodicity argument[1]). We

---

[1] To justify such averaging, we take the oversimplified picture where each atom undergoes Rabi oscillation in the cavity field $b$, with a corresponding Rabi frequency $gb$. If the beam has a spread in velocities $\delta v$, the uncertainty on the Rabi phase of out-coupled atom after a time $1/\Gamma_R$ is given by $gb/\Gamma_R \times \delta v/v$. Given that $gb/\Gamma_R \propto \sqrt{Ng^2/\kappa \Gamma_R} \gg 1$, a relatively small $\delta v/v$ is sufficient to insure that the atoms leave the cavity at a random phase of their Rabi oscillation, which justifies the approximation $\left\langle s_{j-N}^- \right\rangle = \langle S^- \rangle/N$.

expect this to be valid deep in the superradiant regime when the natural timescale for dynamics $Ng^2/\kappa$ greatly exceeds the transit rate $\Gamma_R$. In addition, when $\Gamma_R \gg Ng^2/\kappa$ (in which case no dynamics occurs at all), this approximation holds naturally. This assumption comes in addition to the requirement $Ng^2/\kappa \ll \kappa$ for the adiabatic elimination of $b$ to be valid.

Following a similar approach to estimate the impact on $\langle S^+ \rangle$ and $\langle S^z \rangle$ of a continuous re-loading of atoms in the cavity, and using the fact that variations $\delta S^-_{|load,j}$ occur at a rate $\Gamma$, we find the following set of equations (still with $\hbar = 1$, and $g$, $\kappa$, and $\Gamma$ expressed in $s^{-1}$):

$$
\begin{aligned}
\frac{d\langle S^z \rangle}{dt} &= -4\frac{g^2}{\kappa}\langle S^- \rangle \langle S^+ \rangle + \frac{\Gamma}{2} - \Gamma\frac{\langle S^z \rangle}{N}\,, \\
\frac{d\langle S^- \rangle}{dt} &= 4\frac{g^2}{\kappa}\langle S^z \rangle \langle S^- \rangle - \Gamma\frac{\langle S^- \rangle}{N}\,, \\
\frac{d\langle S^+ \rangle}{dt} &= 4\frac{g^2}{\kappa}\langle S^z \rangle \langle S^+ \rangle - \Gamma\frac{\langle S^+ \rangle}{N}\,.
\end{aligned}
\tag{20}
$$

## 3.2 Superradiant laser in the presence of spontaneous emission

Spontaneous emission into electromagnetic modes that are not the one of the Fabry-Perot cavity can in principle be treated by adding $H_{sp} = \sum_{i,k} g_{sp}\left(s^-_i c^+_k \exp(i\omega_k t) + c_k s^+_i \exp(-i\omega_k t)\right)$ to the Hamiltonian of Eq. (2). Here $c^+_k$ is the creation operator for an electromagnetic mode outside the cavity, with frequency $\omega_k$. $g_{sp}$ is the corresponding coupling strength, that is considered to be constant for all the relevant modes (i.e. those approximately resonant).

The dynamical evolution of the atomic degrees of freedom are then obtained by

$$
i\frac{d\langle s^\epsilon_i \rangle}{dt} = \langle [s^\epsilon_i, H + H_{sp}] \rangle\,,
$$

with $\epsilon = (+, -, z)$. The approach described at the beginning of the paper, with a mean-field description of the external modes $c_k$, predicts the expected decay by spontaneous emission known from the optical Bloch equations for $\langle s^-_i \rangle$ and $\langle s^+_i \rangle$. However, it fails to do so for $\langle s^z_i \rangle$. This failure comes from approximating $\langle s^+_i s^-_i \rangle = \langle s^z_i + 1/2 \rangle$ by $\langle s^+_i \rangle \langle s^-_i \rangle$. The proper result for $[s^z_i, H_{sp}]$ is found by avoiding the early mean-field approximation on the fields $c_k$, and developing the Bloch Langevin equations in the Heisenberg picture, as in [24]. Assuming initially empty external fields, we recover the traditional atomic decays of the Bloch equations. These simply add to the effects described by Eqs. (14), and ultimately, the final equations for the superradiant laser are:

$$
\begin{aligned}
\frac{d\langle S^z \rangle}{dt} &= -4\frac{g^2}{\kappa}\langle S^- \rangle \langle S^+ \rangle - \gamma\left[\langle S^z \rangle + \frac{N}{2}\right] + \frac{\Gamma}{2} - \frac{\Gamma}{N}\langle S^z \rangle\,, \\
\frac{d\langle S^- \rangle}{dt} &= \left[4\frac{g^2}{\kappa}\langle S^z \rangle - \frac{\gamma}{2} - \frac{\Gamma}{N}\right]\langle S^- \rangle\,, \\
\frac{d\langle S^+ \rangle}{dt} &= \left[4\frac{g^2}{\kappa}\langle S^z \rangle - \frac{\gamma}{2} - \frac{\Gamma}{N}\right]\langle S^+ \rangle\,,
\end{aligned}
\tag{21}
$$

where $\gamma$ is the spontaneous emission rate from the excited state. Note that these equations allow for an interpretation of the threshold for a superradiant burst when $\Gamma = 0$. To get a superradiant burst, it is needed that $4\frac{g^2}{\kappa}\langle S^z \rangle - \frac{\gamma}{2} > 0$, so that the initial dipole increases as a function of time. Such a condition also writes

$$
\Delta N \times C > \frac{1}{4}\,,
\tag{22}
$$

where $\Delta N = 2\langle S^z \rangle$ is the population inversion, and $C = \frac{g^2}{\kappa\gamma}$ is the single-atom cavity cooperativity.

# 4 Properties of the continuous superradiant laser at the mean-field level

## 4.1 Synchronisation

To understand why continuous superradiance is possible, we develop the following perturbative argument. We consider $N$ atoms in the cavity at a given time, and we assume that they form a collective dipole corresponding to a macroscopic value of $\langle S^+ \rangle$. We consider that an atom $j$ enters the cavity in its excited state. Similar to Eqs. (10) and (13), the dynamics after this atoms enters is given by:

$$i\frac{d\langle s_j^+ \rangle}{dt} = g\left\langle \left[ s_j^+, s_j^- \right] b^+ \right\rangle = 2g\left\langle s_j^z b^+ \right\rangle \approx 2g\left\langle s_j^z \right\rangle \left\langle b^+ \right\rangle \approx 4i\frac{g^2}{\kappa}\left\langle s_j^z \right\rangle \left\langle S^+ \right\rangle .$$

This shows the main mechanism for synchronisation. An atom with no initial dipole couples to the cavity field, that itself adiabatically follows the pre-existing macroscopic dipole. By this mechanism, the dipole of the incoming atom tends to align with the macroscopic dipole.

## 4.2 From pulsed to continuous superradiance, relaxation oscillations

Solving Eq. (21) allows to understand the main features connecting pulsed and continuous superradiance. We consider an experimental situation where atoms with a velocity perpendicular to the cavity axis $v = 50\,\mathrm{m\,s^{-1}}$ enter this cavity, of transverse mode size $w_0 = 100\,\mu\mathrm{m}$, at a rate $\Gamma$. This corresponds to a refreshing rate – or inverse transit time – $\Gamma_R = \Gamma/N = v/w_0$.

We present in Fig. 2 the solution of these equations for a loading rate $\Gamma$ corresponding to a steady-state atom number $N = 1 \times 10^5$. The other parameters are $g/2\pi = 3\,\mathrm{kHz}$, $\kappa/2\pi = 1000\,\mathrm{kHz}$, and $\gamma/2\pi = 7\,\mathrm{kHz}$ ($C \approx 0.001$). These parameters satisfy $N \times C \gg 1$. Initially, we start from an almost inverted population $\langle S^z \rangle \approx N/2$. In addition, we consider a situation where there initially exists a small but non vanishing dipole $\left( \langle S^- \rangle \approx 0.03\,N \right)$ in order to avoid the steady-state metastable situation inherent to the mean-field approximation.

We observe a train of superradiant light pulses, that relaxes towards a steady-state emission. More precisely, the first superradiant pulse ending up with almost all atoms in the ground state is followed by a slow build-up of the population inversion within the cavity, which eventually leads to a second burst of light. We observe a series of bursts corresponding to peaks in $\langle S^- \rangle$. Through these bursts, the system converges towards a steady state where a constant atomic dipole is sustained in the cavity, corresponding to steady-state lasing. This behaviour is highly reminiscent of relaxation oscillations in standard and superradiant [25] lasers. Interestingly, we find by numerical simulations that the transition to the steady state is achieved in the timescale set by the single-atom transit time $1/\Gamma_R$, irrespective of all other parameters. We have also observed that the larger the number of atoms is, the shorter the first superradiant pulse, as expected from the $Ng^2/\kappa$ scaling for the decay of $\langle S^- \rangle$.

## 4.3 Conditions for continuous superradiance

Since CW superradiance is characterised by a steady state with a macroscopic atomic dipole, this regime can sustain a very small linewidth. To identify the CW regime, we therefore calculate the spectrum of the emitted light using Eq. (4) at long times $t$. In Fig. 3, we plot the measured linewidth when steady state has been reached, for a given number of atoms $N = 100$, and as a function of $\gamma$ and $\Gamma$. Figure 3 clearly defines two regimes. In the region labelled $A$ in Figure 3, *i.e.* for sufficiently small $\Gamma$-dependent $\gamma$ values, continuous superradiance is obtained, characterised by a very small linewidth. In the mean-field equations that we have laid

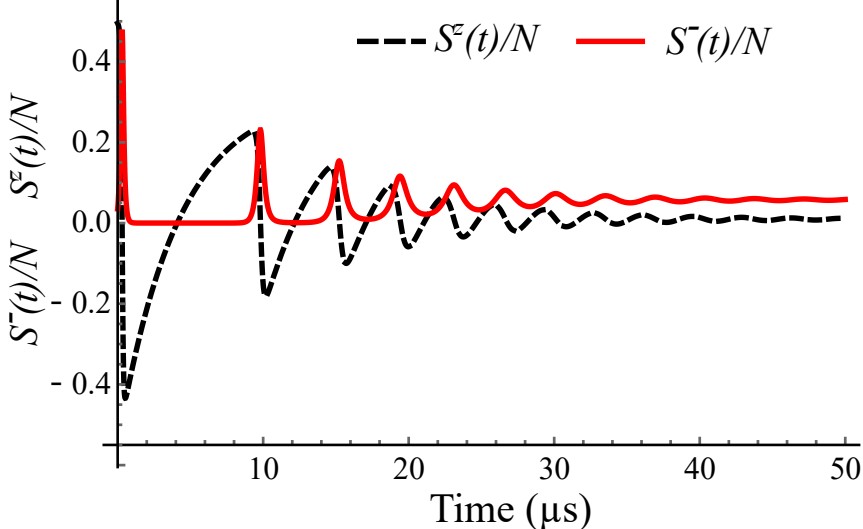

Figure 2: Relaxation to equilibrium for $N = 1 \times 10^5$. We observe a first superradiant burst, followed by a transient dynamics, where subsequent bursts of light drive the system towards steady-state superradiance. We find that those relaxation oscillations are sharper for increasing $N$, and that the timescale for relaxing to the steady state is $1/\Gamma_R$, independent of $N$ and all other parameters.

out, the linewidth can be arbitrarily small for $t \longrightarrow \infty$, which is a pathological outcome of neglecting fluctuations. We will come back to this point at the end of the paper (see Sec. 5.2). In the region labelled $B$ in Figure (3): when $\Gamma$ is small and $\gamma$ large, the linewidth is set by $\gamma$, while when $\gamma$ is small and $\Gamma$ large, the linewidth is roughly $\Gamma/N$, set by the atoms' transit time - this is the single-atom limit where no superradiance occurs; when $\Gamma$ and $\gamma$ are both small, the observed linewidth scales as $N g^2/\kappa$.

We now derive analytical boundaries that separate the continuous superradiant regime to all other regimes. Eqs. (21) allow two families of steady-state solutions (i.e., compatible with $\frac{d\langle S^z \rangle}{dt} = \frac{d\langle S^- \rangle}{dt} = \frac{d\langle S^+ \rangle}{dt} = 0$). One solution is $\langle S^z \rangle = \frac{N}{2}\frac{\Gamma_R - \gamma}{\Gamma_R + \gamma}$, $\langle S^+ \rangle = \langle S^- \rangle = 0$. When $\gamma = 0$, this steady-state situation corresponds to the case where all atoms are in the excited state, with $\langle S^z \rangle = N/2$. This solution is a pathological steady state of mean-field equations. When instead $\Gamma = 0$, the steady-state solution is that of the pulsed regime, where in the end all atoms end up in the ground state and $\langle S^z \rangle = -N/2$.

The second solution is that of the continuous superradiant regime. It corresponds to:

$$\langle S^z \rangle = \frac{1}{8C} + \frac{1}{4NC'}, \tag{23}$$

$$|S^+|^2 = \frac{1}{8C'} - \frac{N}{8C} - \left(\frac{1}{4NC'} + \frac{1}{4C}\right) \times \left(\frac{1}{8C} + \frac{1}{4NC'}\right), \tag{24}$$

where $C = g^2/\kappa\gamma$ is the single atom cavity cooperativity and $C' = g^2/\kappa\Gamma$. The condition for continuous superradiance is that $|S^+|^2 > 0$, i.e.

$$\frac{1}{C} < \frac{1}{2}\left[\frac{-3}{C'N} - 4N + \sqrt{\frac{1}{C'^2 N^2} + \frac{40}{C'} + 16N^2}\right]. \tag{25}$$

The corresponding boundary is shown with the red solid line in Fig. 3. There exists a solution with $C > 0$ only when

$$N^2 C' > \frac{1}{2}. \tag{26}$$

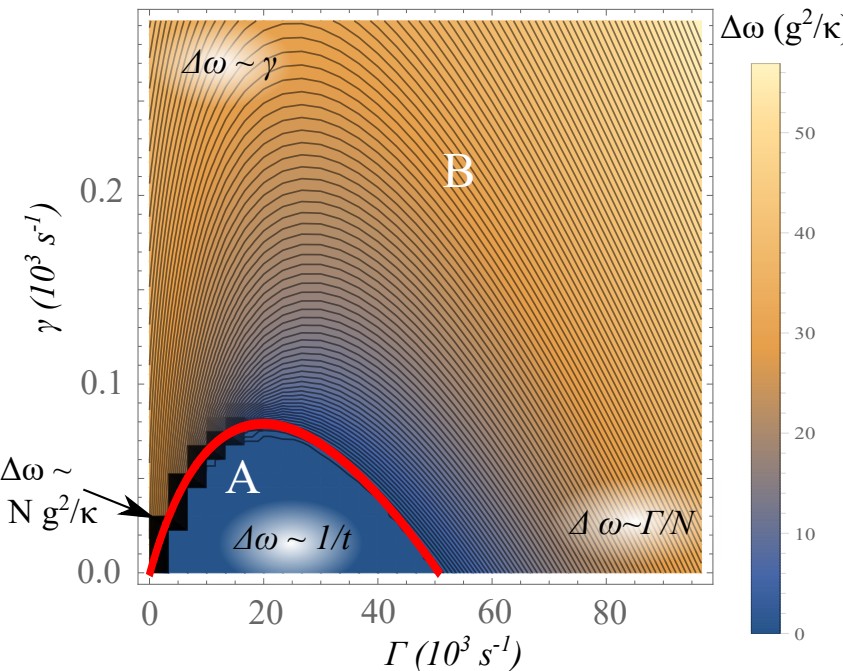

Figure 3: Linewidth $\Delta\omega$ of the emitted light as a function of $\gamma$ and $\Gamma$. The parameters are $g/2\pi = 200$ Hz, $\kappa/2\pi = 1 \times 10^5$ Hz, $N = 100$. Domain $A$ is the CW superradiant emission, characterised by a small linewidth. The red line is the analytical solution from Eq. (25) defining the superradiant threshold. In the domain $B$, light emission is small; we report on the figure the approximate linewidths as a function of the relevant parameters.

This corresponds to $Ng^2/\kappa > \Gamma_R/2$, *i.e.* to the case where the collectively-enhanced emission rate in the cavity is larger than the refreshing rate [14]. A similar condition has been found in the case of a continuously repumped atomic sample [4]. Let us however remind the reader that our model has assumed $Ng^2/\kappa \gg \Gamma_R$ (see Section 3.1). Therefore, this threshold can only be taken as qualitative.

We also note that the condition given by Eq. (25) is a more stringent requirement on the value of $NC$ than that of pulsed superradiance, for any value of $N^2C'$. Indeed, writing this in terms of steady-state inversion population using $\Delta N = 2\langle S^z\rangle$, we find:

$$\Delta N \times C = \frac{1}{4} + \frac{C}{2NC'} > \frac{1}{4} + \frac{1}{-3 - 4C'N^2 + \sqrt{1 + 40C'N^2 + 16C'^2N^4}},$$

from which the pulsed superradiant criterion is recovered when $\Gamma = 0$.

Finally we investigate the behaviour of Eq. (25) at large $N$, which then simplifies to:

$$\gamma < \Gamma_R. \tag{27}$$

This indicates that continuous superradiance is only possible if the finite-time broadening is larger than the spontaneous emission rate. Again, the situation is similar to the continuously repumped case [4]. We also point out that the combined conditions $N^2C' > \frac{1}{2}$ and $\gamma < \Gamma_R$ imply $C \times N > \frac{1}{2}$.

## 4.4 Power of the continuous superradiant laser

We now analyse how the power of the superradiant laser depends on the experimental parameters. From Eq. (24), we have

$$|S^+|^2 = \frac{\kappa}{8g^2}\left((\Gamma_R - \gamma)N - \frac{\gamma^2\kappa}{4g^2} - \frac{\kappa\Gamma_R^2}{2g^2} - \frac{3\gamma\kappa\Gamma_R}{4g^2}\right). \tag{28}$$

In an experiment, the inverse transit time can be tuned by a Zeeman slower and the atom number by the oven temperature. Eq. (28) indicates that the favourable regime is when $\Gamma_R \gg \gamma$, and that, at large $N$ the power is essentially proportional to the number of atoms inside the cavity. This contrasts with pulsed superradiance (at its peak), or with the continuously repumped superradiant laser, for which the emitted power is proportional to $N^2$.

We also estimate the ratio of the number of intracavity photons $N_\nu$ to the number of atoms (using Eqs. (13) and (28)):

$$\frac{N_\nu}{N} = \frac{1}{2N}\left(\frac{N}{\kappa}(\Gamma_R - \gamma) - \frac{\gamma^2}{4g^2} - \frac{\Gamma_R^2}{2g^2} - \frac{3\gamma\Gamma_R}{4g^2}\right). \tag{29}$$

At large $N$, $N_\nu/N \to (\Gamma_R - \gamma)/2\kappa \approx \Gamma_R/2\kappa$. Therefore, when the cavity losses dominate the transit time broadening (as assumed in our theoretical framework), $N_\nu/N \ll 1$, which, following [17], confirms that collective spontaneous emission dominates over stimulated emission.

Finally, we point out that, although the power only linearly scales with $N$, the power of the CW superradiant laser remains in fact large. Indeed, when $N$ is sufficiently large, we can deduce the rate $R$ at which photons exit the cavity

$$R = \kappa N_\nu \approx \frac{N\Gamma_R}{2} = \frac{\Gamma}{2}. \tag{30}$$

This demonstrates that each atom entering the cavity emits on average $1/2$ photon into the mode of the superradiant laser, despite its transit time being much smaller than its natural lifetime. This property, together with the reduced linewidth, is one of the main assets of the superradiant laser, which warrants enough power for applications.

## 4.5 Linewidth in the mean-field approximation

As mentioned above, the mean-field model neglects fluctuations and is therefore unable to account for a CW laser linewidth. Fluctuations originate from loading and unloading of the atoms in the cavity mode, and the associated quantum fluctuations of the atomic dipole. Here, we use a Monte-Carlo approach where the loading and unloading of atoms is explicitly added to the mean-field equations. We then compare our numerical results for the linewidth to an analytical estimate based on a phase diffusion.

### 4.5.1 Monte-Carlo approach

We use the equations derived above for the coupled atom-cavity dynamics (Eqs. (14), *i.e.* without spontaneous emission), and replace the description of the average effect of loading of atoms in the cavity by a stochastic process. Namely, we consider that each newly loaded atom has $\langle s_z \rangle = 1/2$, whereas $\langle s_x \rangle$ and $\langle s_y \rangle$ randomly take the values $1/2$ or $-1/2$ [19]. Each atom $j$ loaded at time $t_j = j/\Gamma$ instantaneously modifies the collective atomic dipole, and the subsequent dynamics is simulated using Eqs. (14) until another atom enters the cavity. The disappearance of an atom from the cavity is described by a discrete reduction of the average atomic operators: $S^\epsilon \to (N-1)/N \times S^\epsilon$ after dynamics has taken place. In addition, we add a stochastic noise corresponding to the quantum noise of a spin $1/2$ whose direction is the direction of the collective spin when the atom exits the cavity. This procedure (loading, dynamical evolution, unloading) is repeated many times to describe the successive loading of many atoms into the cavity. We finally calculate the atomic dipole autocorrelation function

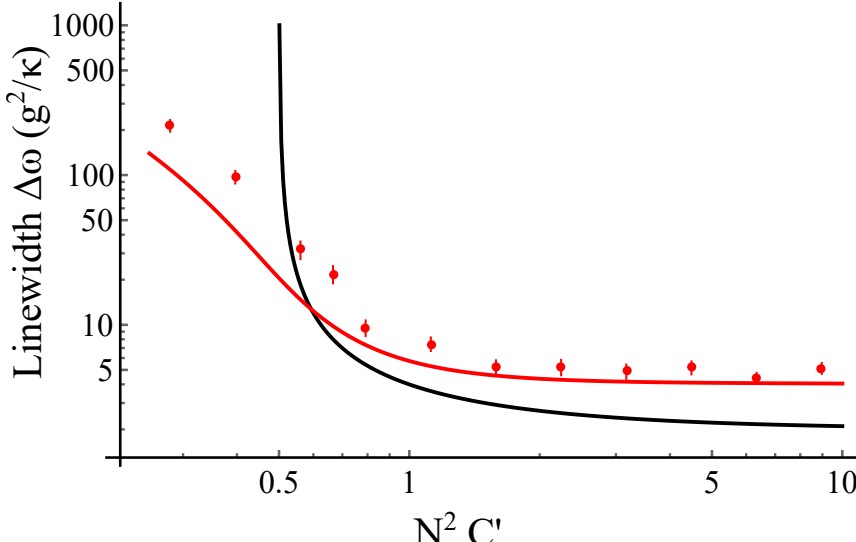

Figure 4: Linewidth estimate in the mean-field approximation. We plot the results of our Monte-Carlo simulation (in the mean-field approximation, red bullets), together with the analytical estimate from Eq. (31) (black solid line), as a function of $N^2C'$. Calculations are for 100 atoms, and the Monte-Carlo results are averaged over 50 realisations (the corresponding error bars are standard deviations derived from the fitted autocorrelation time). The divergence on the left side of the figure is the limit $N^2C' = 1/2$. The red solid line is the result of the linewidth from the cumulant expansion, see Eq. (49). For this figure parameters are $g/2\pi = 300$ Hz, $\kappa/2\pi = 1 \times 10^5$ Hz.

$C_S(\Delta t) = \langle S_x(t + \Delta t)S_x(t)\rangle$, and fit it by an exponential decay of the form $\exp(-\Delta t/\tau_c)$. From this fit we deduce the correlation time $\tau_c$ and the laser pulsation linewidth half-width-half-maximum (HWHM) $\Delta\omega = 2\pi\Delta\nu = 1/\tau_c = D/2$ (where $D$ is the phase drift coefficient, see below). This approach is similar to the one used in [14]. Results are shown in Fig. 4 together with the analytical estimates that we will now discuss. The linewidth that is represented is the HWHM pulsation width scaled to $g^2/\kappa$ (in $s^{-1}$).

### 4.5.2 A heuristic approach for the linewidth estimate

Another common way to calculate the linewidth is to estimate the phase drift of the light [26, 27], see[2]. In our adiabatic approximation, the phase of the laser is locked to that of the atomic dipole, and we therefore will estimate the phase drift from the fluctuations of the collective atomic dipole.

Without loss of generality, we take the case where the collective dipole points in the $x$ direction, setting $\langle S^-\rangle = \langle S^+\rangle$. When an atom leaves the cavity, which happens at a rate $\Gamma$, its spin in the horizontal plane is effectively measured, and a back action on the collective spin introduces a random walk in the collective spin direction. Given that the mean-field approximation assumes that particles are uncorrelated, the random walk step in the direction with zero mean dipole is $\delta S_y^2 = 1/4$. The same can be said for the atoms entering the cavity.

---

[2] A way to explicitly relate the phase drift to the width of the laser is to connect it to the field autocorrelation. We assume that the laser field reads: $E(t) = \cos(\phi(t))$, and that the phase drift is characterised by $\langle(\phi(t + \Delta t) - \phi(t))^2\rangle \equiv D\Delta t$ where $D$ is the drift coefficient. We then have: $C_E(\Delta t) = \langle E(t + \Delta t)E(t)\rangle \approx \cos(\phi(t + \Delta t) - \phi(t))$. Using a small-time expansion, we have $C_E(\Delta t) \approx 1 - \frac{1}{2}\langle(\phi(t + \Delta t) - \phi(t))^2\rangle$. The correlation time $\tau_c$ is related to the phase drift coefficient $D$ by setting $C_E(\Delta t) \propto \exp(-\Delta t/\tau_c)$, so that $1/\tau_c = D/2$.

Taken together, these processes are the main mechanism for phase diffusion, which sets the laser linewidth.

We take the large atom limit, assume that the dipole is large enough and thus neglect the dipole length fluctuation. In this regime, we expect a drift for the phase of the laser that is given by the drift coefficient:

$$D = \Gamma \frac{2\,\delta S_y^2}{|S_x|^2} = \frac{\Gamma}{2} \frac{1}{|S_x|^2}\,.$$

In the large N limit

$$D = \frac{4g^2}{\kappa} \left[ \frac{N^2 C'}{N^2 C' - \frac{1}{2}} \right],\tag{31}$$

and the corresponding laser linewidth (half-width-half-maximum) $\Delta\omega = D/2$.

We compare both analytical and numerical approaches in Figure 4. Both approaches are in qualitative agreement, as they show a similar scale for the ultimate linewidth set by the Purcell rate $\propto g^2/\kappa$ deep in the superradiant regime, and a degradation of the linewidth when $N^2 C'$ approaches $1/2$. We point out that the analytical model can only describe the laser linewidth deep in the superradiant lasing regime, since it assumes both that the collective spin is large, and that $\Gamma_R \ll N g^2/\kappa$ (such that a steady state is obtained for each atom after it is injected in the cavity). We also note a qualitative disagreement between both approaches when $N^2 C'$ approaches $1/2$. In particular, the divergence in the analytical model is pathological and does not occur in the Monte-Carlo simulations. At small $\Gamma_R$, where $N^2 C' \gg 1$, we find that both our methods are in qualitative agreement with those of [14]. Finally, we also point out the good qualitative agreement between the linewidth deduced from Monte-Carlo simulations and the linewidth, given by Eq. (49), calculated using the second order cumulant expansion approach that we will now describe.

# 5 Correlations and linewidth in the continuous superradiant regime

Atom-atom correlations, neglected so far, can have a strong impact on the fluctuations of a collective dipole (see for example spin squeezing [28]). The purpose of this section is to take into account these correlations, within a second-order cumulant expansion, and estimate their impact on the CW superradiant laser linewidth. For simplicity, and since the relevant regime is $\Gamma_R \gg \gamma$, we neglect spontaneous emission in this section.

## 5.1 Equations for correlators

To go beyond the mean-field approximation, we resort to the quantum version of the Langevin equation. For a given operator Q, time-dependent in the Heisenberg picture, this reads [20]:

$$\frac{dQ}{dt} = -i[Q, H_0] - \frac{\kappa}{2}[Q, b^+]b + \frac{\kappa}{2}b^+[Q, b] + f_Q(t).\tag{32}$$

This equation relies on the same approximation made earlier (that led to Eq. 6). In our model, the only stochastic process is the out-coupling of photons from the cavity (which does not affect atomic observables). As before, assuming initially empty external modes, the stochastic term $f_b$ and $f_{b^+}$ have no impact on all first- and second-order operators used below. We thus set $f_Q = 0$.

As shown above (see *e.g.* Eq. (10)), the time derivative of the average of single-atom spin operators generally involves averages of a product of operators, for example $i\frac{d\langle s_i^-\rangle}{dt} = -2g\langle s_i^z b\rangle$. To go beyond a mean-field description, we therefore need to calculate the time dependence of quantum averages of product of operators such as $\langle s_i^z b\rangle$. For example,

$$\frac{ds_1^z b}{dt} = -\frac{\kappa}{2}s_1^z b - igs_1^z\sum_{j\neq 1}s_j^- - ig\left(-b^+ bs_1^- + s_1^z s_1^- + b^2 s_1^+\right),\qquad(33)$$

which involves products of three operators. Our approach is to derive a close set of equations connecting first- and second-order operators only, in a so-called second-order cumulant expansion [16]. The first correction to the mean-field approximation (in which one neglects the second order cumulant such that $\langle X_1 X_2\rangle = \langle X_1\rangle\langle X_2\rangle$ for two operators $X_1$ and $X_2$) is indeed to neglect the third order cumulant, thus postulating [29]

$$\langle X_1 X_2 X_3\rangle = \langle X_1 X_2\rangle\langle X_3\rangle + \langle X_2 X_3\rangle\langle X_1\rangle + \langle X_3 X_1\rangle\langle X_2\rangle - 2\langle X_1\rangle\langle X_2\rangle\langle X_3\rangle.\qquad(34)$$

We also take into account the effect of loading and unloading of atoms. For this, we define:

$$\Sigma^{(\epsilon,\mu)} = \frac{1}{2}\sum_{i<j}\left(s_i^\epsilon(t)s_j^\mu(t) + s_i^\mu(t)s_j^\epsilon(t)\right).\qquad(35)$$

We follow the same procedure as described above for the effect of loading and unloading on the first moments of the atomic parameters, to find the equation on products of operators:

$$\frac{d\langle\Sigma^{(\epsilon,\mu)}\rangle}{dt}\bigg|_{load} = \sum_{i<j}\left(\frac{d\eta_i(t)}{dt}\eta_j + \frac{d\eta_j(t)}{dt}\eta_i\right)\left(\frac{\langle s_i^\epsilon s_j^\mu\rangle + \langle s_i^\mu s_j^\epsilon\rangle}{2}\right).\qquad(36)$$

For a given duration $1/\Gamma$ we only consider the time interval $\left[t_{i_0}, t_{i_0}+\tau_0\right]$ where atom $i_0$ enters the cavity and atom $i_0-N$ leaves the cavity, such that during this duration $\frac{d\eta_{i_0}(t)}{dt} = 1/\tau_0$ and $\frac{d\eta_{i_0-N}(t)}{dt} = -1/\tau_0$ (the derivatives are zero otherwise). We thus find:

$$\frac{d\langle\Sigma^{(\epsilon,\mu)}\rangle}{dt}\bigg|_{load} \approx -\frac{\Gamma}{2}\sum_{i_0-N<j}\left(\langle s_{i_0-N}^\epsilon s_j^\mu\rangle + \langle s_{i_0-N}^\mu s_j^\epsilon\rangle\right) + \frac{\Gamma}{2}\sum_{i<i_0}\left(\langle s_i^\epsilon s_{i_0}^\mu\rangle + \langle s_i^\mu s_{i_0}^\epsilon\rangle\right).\qquad(37)$$

The atoms entering the cavity are uncorrelated with the atoms inside the cavity when entering, so that $\langle s_i^\epsilon s_{i_0}^\mu\rangle \approx \langle s_i^\epsilon\rangle\langle s_{i_0}^\mu\rangle$. Here, $\langle s_{i_0}^\mu\rangle = s_0^\mu$ is the value of the spin operator for an incoming atom. We have $\sum_{i<i_0}\langle s_i^\epsilon\rangle \approx S^\epsilon$.

On the other hand, we will further assume that $\frac{1}{2}\sum_{i_0-N<j}\left(\langle s_{i_0-N}^\epsilon s_j^\mu\rangle + \langle s_{i_0-N}^\mu s_j^\epsilon\rangle\right) \approx \frac{2}{N-1}\Sigma^{(\epsilon,\mu)}$ (i.e. the pair of atom $(i_0-N, j)$ is representative of the average correlator.) We find the loading-unloading terms:

$$\frac{d\langle\Sigma^{(\epsilon,\mu)}\rangle}{dt}\bigg|_{load} \approx -\frac{2\Gamma}{N-1}\langle\Sigma^{(\epsilon,\mu)}\rangle + \frac{\Gamma}{2}\left(\langle S^\epsilon\rangle s_0^\mu + \langle S^\mu\rangle s_0^\epsilon\right).\qquad(38)$$

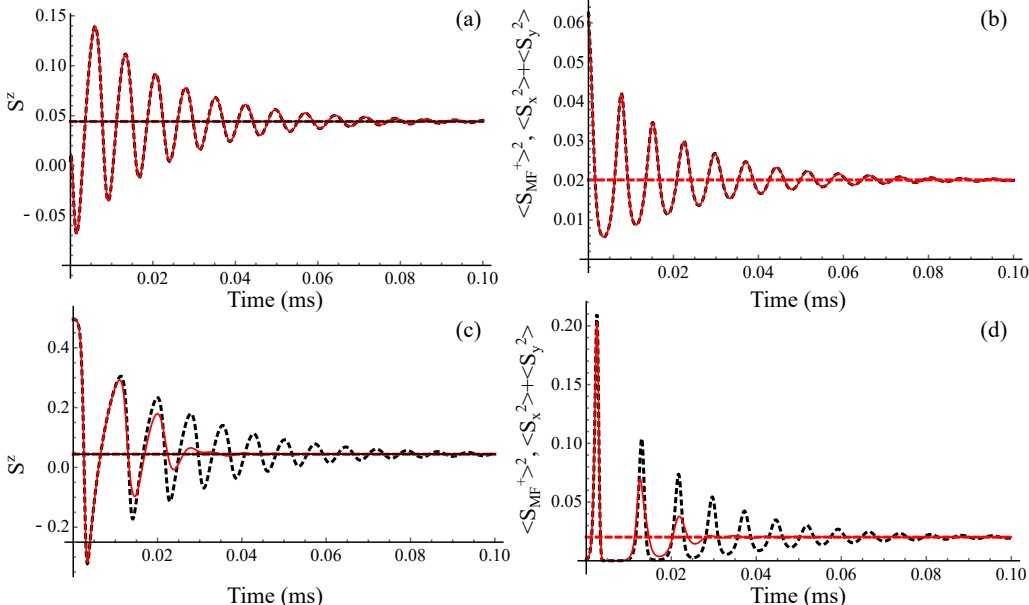

Figure 5: Comparison between mean-field dynamics (dashed, black) and the cumulant approximation method (solid red). Calculations for $g/2\pi = 3000\,\text{Hz}$, $\kappa/2\pi = 1 \times 10^6\,\text{Hz}$, $\Gamma_R/2\pi = 2 \times 10^5\,\text{Hz}$, and $N = 2 \times 10^4$. (a) and (b) correspond to an initial macroscopic dipole, while (c) and (d) correspond to a small initial atomic dipole. Left ((a) and (c)): $S^z$; Right ((b) and (d)): $\left\langle S_{MF}^+ \right\rangle^2$, and its quantum counterpart $\left\langle S_x^2 + S_y^2 \right\rangle$. The horizontal lines are the analytical asymptotic values.

These terms need to be added to those corresponding to the cumulant expansion, in order to find the final equations describing the dynamics of the system in presence of the loading term. We finally assume that all atoms in the cavity mode have identical properties, *i.e.* $\forall (i,j), \left\langle s_i^\epsilon s_j^\mu \right\rangle = \left\langle s_1^\epsilon s_2^\mu \right\rangle, \left\langle s_i^\mu \right\rangle = \left\langle s_1^\mu \right\rangle$. We find:

$$\frac{d\left\langle s_j^- \right\rangle}{dt} = 2ig \left\langle s_j^z b \right\rangle - \frac{\Gamma}{N} \left\langle s_j^- \right\rangle,$$

$$\frac{d\left\langle s_j^z \right\rangle}{dt} = ig \left\langle s_j^- b^+ \right\rangle - ig \left\langle s_j^+ b \right\rangle - \frac{\Gamma}{N} \left\langle s_j^z \right\rangle + \frac{\Gamma}{2N},$$

$$\frac{d\left\langle b \right\rangle}{dt} = -\frac{\kappa}{2} \left\langle b \right\rangle - ig \sum_j \left\langle s_j^- \right\rangle,$$

$$\frac{d\left\langle b^+ b \right\rangle}{dt} = -\kappa \left\langle b^+ b \right\rangle - ig \sum_j \left\langle s_j^- b^+ - s_j^+ b \right\rangle,$$

$$\frac{d\left\langle b^2 \right\rangle}{dt} = -\kappa \left\langle b^2 \right\rangle - 2ig \sum_j \left\langle s_j^- b \right\rangle,$$

$$\begin{aligned}
\frac{d\left\langle s_1^z b \right\rangle}{dt} = &-\frac{\Gamma}{N} \left\langle s_1^z b \right\rangle + \frac{\Gamma}{2N} \left\langle b \right\rangle - \frac{\kappa}{2} \left\langle s_1^z b \right\rangle - ig(N-1)\left\langle s_1^z s_2^- \right\rangle + \frac{i}{2}g \left\langle s_1^- \right\rangle \\
&+ ig\Big(\left\langle b^+ \right\rangle\left\langle b s_1^- \right\rangle + \left\langle b \right\rangle\left\langle b^+ s_1^- \right\rangle + \left\langle s_1^- \right\rangle\left\langle b^+ b \right\rangle - 2\left\langle b \right\rangle\left\langle b^+ \right\rangle\left\langle s_1^- \right\rangle - 2\left\langle b s_1^+ \right\rangle\left\langle b \right\rangle \\
&- \left\langle s_1^+ \right\rangle\left\langle b^2 \right\rangle + 2\left\langle b \right\rangle^2 \left\langle s_1^+ \right\rangle\Big),
\end{aligned}$$

$$\frac{d\left\langle b^+ s_1^-\right\rangle}{dt} = -\frac{\Gamma}{N}\left\langle s_1^- b^+\right\rangle - \frac{\kappa}{2}\left\langle b^+ s_1^-\right\rangle + ig(N-1)\left\langle s_1^- s_2^+\right\rangle + ig\left(\left\langle s_1^z\right\rangle + \frac{1}{2}\right) \tag{39}$$
$$+ 2ig\left(\left\langle b^+ b\right\rangle\left\langle s_1^z\right\rangle + \left\langle b s_1^z\right\rangle\left\langle b^+\right\rangle + \left\langle b^+ s_1^z\right\rangle\left\langle b\right\rangle - 2\left\langle b\right\rangle\left\langle b^+\right\rangle\left\langle s_1^z\right\rangle\right),$$

$$\frac{d\left\langle b s_1^-\right\rangle}{dt} = -\frac{\Gamma}{N}\left\langle s_1^- b\right\rangle - \frac{\kappa}{2}\left\langle b s_1^-\right\rangle - ig(N-1)\left\langle s_1^- s_2^-\right\rangle$$
$$+ 2ig\left(\left\langle b^2\right\rangle\left\langle s_1^z\right\rangle + 2\left\langle b s_1^z\right\rangle\left\langle b\right\rangle - 2\left\langle b\right\rangle^2\left\langle s_1^z\right\rangle\right),$$

$$\frac{d\left\langle s_1^+ s_2^-\right\rangle}{dt} = -2\frac{\Gamma}{N-1}\left\langle s_1^+ s_2^-\right\rangle - 2ig\left(\left\langle b^+ s_2^-\right\rangle\left\langle s_1^z\right\rangle + \left\langle b^+ s_1^z\right\rangle\left\langle s_2^-\right\rangle + \left\langle s_1^z s_2^-\right\rangle\left\langle b^+\right\rangle\right.$$
$$\left. - 2\left\langle b^+\right\rangle\left\langle s_2^-\right\rangle\left\langle s_1^z\right\rangle - \left\langle b s_2^z\right\rangle\left\langle s_1^+\right\rangle - \left\langle b s_1^+\right\rangle\left\langle s_2^z\right\rangle - \left\langle s_2^z s_1^+\right\rangle\left\langle b\right\rangle + 2\left\langle b\right\rangle\left\langle s_2^z\right\rangle\left\langle s_1^+\right\rangle\right),$$

$$\frac{d\left\langle s_1^+ s_2^+\right\rangle}{dt} = -2\frac{\Gamma}{N-1}\left\langle s_1^+ s_2^+\right\rangle - 2ig\left(\left\langle b^+ s_2^+\right\rangle\left\langle s_1^z\right\rangle + \left\langle b^+ s_1^z\right\rangle\left\langle s_2^+\right\rangle + \left\langle s_1^z s_2^+\right\rangle\left\langle b^+\right\rangle\right.$$
$$\left. - 2\left\langle b^+\right\rangle\left\langle s_2^+\right\rangle\left\langle s_1^z\right\rangle + \left\langle b^+ s_2^z\right\rangle\left\langle s_1^+\right\rangle + \left\langle b^+ s_1^+\right\rangle\left\langle s_2^z\right\rangle + \left\langle s_2^z s_1^+\right\rangle\left\langle b^+\right\rangle\right.$$
$$\left. - 2\left\langle b^+\right\rangle\left\langle s_2^z\right\rangle\left\langle s_1^+\right\rangle\right),$$

$$\frac{d\left\langle s_1^z s_2^-\right\rangle}{dt} = -2\frac{\Gamma}{N-1}\left\langle s_1^z s_2^-\right\rangle + \frac{\Gamma}{2(N-1)}\left\langle s_1^-\right\rangle - ig\left(\left\langle s_1^+ s_2^-\right\rangle\left\langle b\right\rangle + \left\langle s_1^+ b\right\rangle\left\langle s_2^-\right\rangle + \left\langle s_2^- b\right\rangle\left\langle s_1^+\right\rangle\right.$$
$$\left. - 2\left\langle s_1^+\right\rangle\left\langle s_2^-\right\rangle\left\langle b\right\rangle - \left\langle s_1^- s_2^-\right\rangle\left\langle b^+\right\rangle - \left\langle s_1^- b^+\right\rangle\left\langle s_2^-\right\rangle - \left\langle s_2^- b^+\right\rangle\left\langle s_1^-\right\rangle + 2\left\langle s_1^-\right\rangle\left\langle s_2^-\right\rangle\left\langle b^+\right\rangle\right.$$
$$\left. - 2\left\langle s_1^z s_2^z\right\rangle\left\langle b\right\rangle - 2\left\langle s_1^z b\right\rangle\left\langle s_2^z\right\rangle - 2\left\langle s_2^z b\right\rangle\left\langle s_1^z\right\rangle + 4\left\langle s_1^z\right\rangle\left\langle s_2^z\right\rangle\left\langle b\right\rangle\right),$$

$$\frac{d\left\langle s_1^z s_2^z\right\rangle}{dt} = -2\frac{\Gamma}{N-1}\left\langle s_1^z s_2^z\right\rangle + \frac{\Gamma}{N-1}\left\langle s_1^z\right\rangle - ig\left(\left\langle s_1^+ s_2^z\right\rangle\left\langle b\right\rangle + \left\langle s_1^+ b\right\rangle\left\langle s_2^z\right\rangle + \left\langle s_2^z b\right\rangle\left\langle s_1^+\right\rangle\right.$$
$$\left. - 2\left\langle s_1^+\right\rangle\left\langle s_2^z\right\rangle\left\langle b\right\rangle - \left\langle s_1^- s_2^z\right\rangle\left\langle b^+\right\rangle - \left\langle s_1^- b^+\right\rangle\left\langle s_2^z\right\rangle - \left\langle s_2^z b^+\right\rangle\left\langle s_1^-\right\rangle + 2\left\langle s_1^-\right\rangle\left\langle s_2^z\right\rangle\left\langle b^+\right\rangle\right.$$
$$\left. + \left\langle s_1^z s_2^+\right\rangle\left\langle b\right\rangle + \left\langle s_1^z b\right\rangle\left\langle s_2^+\right\rangle + \left\langle s_2^+ b\right\rangle\left\langle s_1^z\right\rangle - 2\left\langle s_1^z\right\rangle\left\langle s_2^+\right\rangle\left\langle b\right\rangle - \left\langle s_1^z s_2^-\right\rangle\left\langle b^+\right\rangle\right.$$
$$\left. - \left\langle s_1^z b^+\right\rangle\left\langle s_2^-\right\rangle - \left\langle s_2^- b^+\right\rangle\left\langle s_1^z\right\rangle + 2\left\langle s_1^z\right\rangle\left\langle s_2^-\right\rangle\left\langle b^+\right\rangle\right).$$

We first look for stationary solutions to Eq. (39). In steady state, the collective dipole will point in a given direction, which, by rotational symmetry, can be taken arbitrarily, *e.g.* along the $x$ direction. Therefore, without loss of generality, we postulate $\left\langle s_j^+\right\rangle = \left\langle s_j^-\right\rangle$, which implies:

$\left\langle s_j^+ s_k^+\right\rangle = \left\langle s_j^- s_k^-\right\rangle, \left\langle s_j^z s_k^+\right\rangle = \left\langle s_j^z s_k^-\right\rangle, \left\langle b^+\right\rangle = -\left\langle b^-\right\rangle, \left\langle s_j^z b^+\right\rangle = -\left\langle s_j^z b^-\right\rangle, \left\langle s_j^+ b^+\right\rangle = -\left\langle s_j^- b^-\right\rangle.$

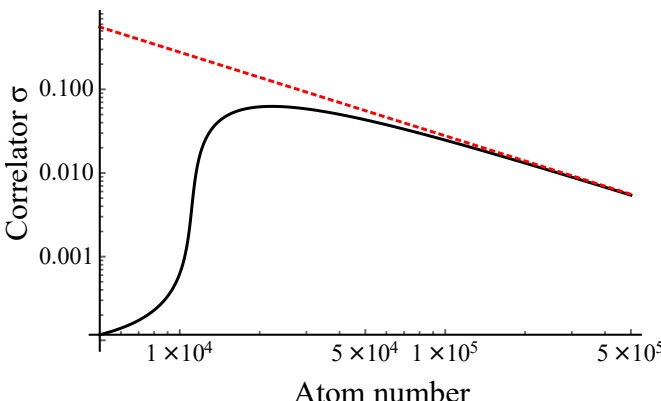

Figure 6: Steady-state correlator $\sigma$, calculated for $g/2\pi = 3000\,\text{Hz}$, $\kappa/2\pi = 1 \times 10^6\,\text{Hz}$, $\Gamma_R/2\pi = 2 \times 10^5\,\text{Hz}$ as a function of atom number. The dashed line is Eq. (43). The correlator rapidly increases above the threshold for superradiance, but scales as $1/N$ at large atom number.

We find the following stationary solutions for the two-body atomic operators (defining $r = \Gamma_R/\kappa$): $\langle s^+ \rangle = \langle s_1^+ s_2^+ \rangle = \langle s_1^z s_2^+ \rangle = 0$, and

$$
\begin{aligned}
s^z \equiv \langle s^z \rangle = {} & \frac{1 + 2r + 2C'(1 + N^2(1 + 2r))}{8C'(1 + N(N + 2Nr - 2))} \\
& - \frac{\sqrt{(2C' + (1 + 2C'N^2)(1 + 2r))^2 + 8C'(-1 + 4C'N - 2r)(1 + N(N + 2Nr - 2))}}{8C'(1 + N(N + 2Nr - 2))}, \\
\sigma \equiv \langle s_1^- s_2^+ \rangle = {} & \frac{1 - 4C'N + 2r - 2(1 + 2r + 4C'N(1 + Nr))s^z + 16C'N^2 r s^{z\,2}}{8C'(N - 1)N}, \\
\langle s_1^z s_2^z \rangle = {} & \frac{s^z + 2(N - 1)s^{z\,2}}{2N},
\end{aligned}
\tag{40}
$$

where $\sigma$ is a two-body correlator that will impact the width of the laser (see Section 5.2). For the field operators, we find $\langle b \rangle = \langle b^2 \rangle = \langle s^- b \rangle = \langle s^z b \rangle = 0$, and

$$
\begin{aligned}
\langle b^+ b \rangle &= \frac{\Gamma(1 - 2s^z)}{2\kappa}, \\
\langle s^- b^+ \rangle &= \frac{\Gamma_R(s^z - 1/2)}{2ig}.
\end{aligned}
\tag{41}
$$

One key difference with the mean-field results is the vanishing value of $\langle s^+ \rangle$, which arises due to the phase invariance of the system in the equatorial plane [4]. Namely, the classical dipole points in a random direction, which leads to $\langle s^+ \rangle = 0$. However, one can still define the length of the collective atomic dipole using its variance $\langle S_x^2 + S_y^2 \rangle$. We compare the steady-state solutions within the mean-field approximation, $(S_{MF}^+)^2$, given by Eq. (24), to the results of the cumulant expansion for $\langle S_x^2 + S_y^2 \rangle$. We find, for large atom number, at first order in $1/N$:

$$
\frac{\langle S_x^2 + S_y^2 \rangle}{(S_{MF}^+)^2} = \frac{8g^2 + 2\Gamma_R\kappa + \kappa^2}{2\Gamma_R\kappa + \kappa^2} + \frac{(4g^2 + \Gamma_R k)(4g^2 - \Gamma_R(2\Gamma_R + \kappa))}{2g^2\Gamma_R(2\Gamma_R + \kappa)}\frac{1}{N},
\tag{42}
$$

which approaches 1 in the limit of large $N$ and $g \ll (\kappa, \Gamma_R)$. Likewise, we can compare the values for steady-state magnetisation in the mean-field and cumulant approaches, and find a very good agreement when the atom number is large.

We show in Fig. 5 the results of our numerical dynamical simulations for two different approximations: the mean-field approximation (which we here solve without the adiabatic approximation on $b$), and the second-order cumulant approach. In the figure, both curves are almost superimposed when the dynamics starts with a macroscopic dipole. The steady-state limits, also shown, are almost identical. We nevertheless find that the mean-field approximation and the cumulant approach can lead to significant dynamical differences when the initial dipole is small. In particular, in that case, the oscillation relaxations are damped faster in the cumulant approximation, as compared to the mean-field approximation. The steady-state values however almost exactly coincide.

Finally, we also plot in Fig. 6 the steady-state correlator $\sigma = \langle s_1^+ s_2^- \rangle$ as a function of the number of atoms. In the large $N$ limit, we find:

$$
\sigma \approx \frac{\kappa\left(\frac{\Gamma_R}{g^2} - \frac{4}{2\Gamma_R + \kappa}\right)}{8N}.
\tag{43}
$$

Correlations are found to be maximum slightly above the threshold for superradiant lasing, and algebraically decay back to zero at large $N$. We discuss in the next section how $\sigma$ can be related to the laser linewidth.

## 5.2 Fluctuation of the collective dipole, correlations, and laser linewidth

### 5.2.1 Linewidth calculation

To derive the laser linewidth, we proceed by calculating the autocorrelation function:

$$C_S(dt) = \langle S_x(t+dt)S_x(t)\rangle \,, \tag{44}$$

where the average stands both for the quantum average and the temporal average over $t$. We have:

$$i\frac{dS^-}{dt} = -2gS^z b - i\frac{\Gamma}{N}S^- \,, \tag{45}$$

which we expand at first order in $dt$: $S^-(t+dt) = S^-(t) - \frac{2gS^z b}{i}dt - \frac{\Gamma}{N}S^- dt$. We then find, at first order in $dt$:

$$\langle S_x(t+dt)S_x(t)\rangle = \langle S_x(t)^2\rangle + \frac{1}{4}\left\langle \frac{2gS^z b^+}{i}S^- dt - \frac{2gS^z b}{i}S^+ dt - \frac{\Gamma dt}{N}\left(S^+ S^- + S^- S^+\right)\right\rangle \,. \tag{46}$$

We again use the second order cumulant expansion, which, in the steady state for which $\langle b\rangle = \langle s^+\rangle = 0$, implies that $\langle s^z b^+ s^-\rangle = \langle b^+ s_-\rangle\langle s^z\rangle$. Assuming that

$$\langle S_x(t+dt)S_x(t)\rangle = \langle S_x^2\rangle \exp\left(-\frac{dt}{\tau_c}\right) \,, \tag{47}$$

we find:

$$\frac{1}{\tau_c} = \frac{\Gamma_R}{\langle S_x^2\rangle}\left(\langle S_x^2\rangle + \frac{N^2}{2}s^z\left(s^z - \frac{1}{2}\right)\right) \,, \tag{48}$$

which can also be written as:

$$\frac{1}{\tau_c} = \frac{\Gamma_R}{\langle S_x^2\rangle}\left(\frac{N}{4} + \frac{N(N-1)}{2}\sigma + \frac{N^2}{2}s^z\left(s^z - \frac{1}{2}\right)\right) \,, \tag{49}$$

where we have used $\langle S_x^2\rangle = \frac{N}{4} + \frac{1}{4}\sum_{i\neq j}\langle s_i^+ s_j^+ + s_i^+ s_j^- + s_i^- s_j^+ + s_i^- s_j^-\rangle = \frac{N}{4} + \frac{1}{2}N(N-1)\sigma$ (in the steady-state regime).

Eq. (49) provides an analytical expression for the laser linewidth (since $s^z$ and $\sigma$ have been given in Eq. (40)). Performing a large $N$ expansion of the term inside the parenthesis of Eq. (49) provides interesting insight on this final result. For $N^2 C' \gg 1$ we find:

$$\frac{1}{\tau_c} \approx \frac{\Gamma}{2L^2}(1-4\sigma) \,, \tag{50}$$

where $L = \sqrt{\langle S_x^2 + S_y^2\rangle}$ is the dipole length. As mentioned above, $\tau_c$ is related to the drift coefficient and laser linewidth HWHM using $\Delta\omega = 1/\tau_c = D/2$. In the large atom limit, since $\langle S_x^2\rangle = \frac{N}{4} + \frac{1}{2}N(N-1)\sigma$, Eq. (50) shows that the laser linewidth is defined by $\sigma$ and $N$ only. The minimum linewidth HWHM is $\Delta\omega = 4g^2/\kappa$, obtained when $N^2 C' \gg 1$.

### 5.2.2 Interpretation and discussion

The above derivation illustrates the transition from a collection of independent atoms to a collective object: in Eq. (48), the first term in the parenthesis (that dominates when all atoms remain in the excited state, with $s^z = 1/2$) directly reflects the replacement of all the atoms at the timescale $1/\Gamma_R$. The second term, resulting from atom-cavity interactions, is negative and, in the superradiant regime, compensates for most of the first, leading to Eq. (50).

This latter equation is suggestive of the following physical interpretation. It corresponds to a long-lived collective spin of length $L$, that suffers perturbations due to the single-atom shot noise at the rate $\Gamma$. The first term in the parenthesis of Eq. (50) hence corresponds to the variance of the spin of the out-and-in-coupled atoms. This is equivalent to what we have seen when describing the laser linewidth in the mean-field model. However, Eq. (50) also takes into account the effect of atom-atom correlations associated with $\sigma$ on the linewidth.

To qualitatively explain the effect of two-body correlations, we propose the following argumentation. For this we define a phase $\delta_i \equiv s_i^y/L$, that corresponds to the rotation in the $xy$ plane of a collective classical spin of length $L$ initially pointing in the $x$ direction, associated with the measurement of the spin of atom $i$ in the $y$ direction. To take into account correlations, we need to describe the effect of outcoupling two consecutive atoms. What matters to estimate the phase drift is the variance of $\delta_i - \delta_{i+1}$ for a series of pairs exiting the cavity at a rate $\Gamma/2$. Therefore the phase drift due to out-coupling will be given by a coefficient:

$$D' = \frac{\Gamma}{2}(\text{Var}(\delta_i - \delta_{i+1})) = \frac{\Gamma}{2}(\text{Var}(\delta_i) + \text{Var}(\delta_{i+1}) - 2\text{CoVar}(\delta_i, \delta_{i+1}))$$
$$= \frac{\Gamma}{L^2}\left(\frac{1}{4} - \text{CoVar}(s_i^y, s_{i+1}^y)\right),$$
(51)

where Var and CoVar respectively stand for the variance and the covariance. In the steady state $\text{CoVar}(s_i^y, s_{i+1}^y) = \frac{1}{2}\langle s_1^+ s_2^- \rangle = \frac{\sigma}{2}$, so that the drift is given by $D' = \frac{\Gamma}{4L^2}(1 - 2\sigma)$. Finally, we also need to take into account the phase drift associated with the insertion of atoms. This drift adds to the out-coupling term, and corresponds to un-correlated atoms, therefore the total drift is: $D'' = \frac{\Gamma}{2L^2}(1 - \sigma)$. Note that, other than the description of correlations between out-coupled atoms, this approach is identical to that presented in Section 4.5.2 and we indeed find the same result when $\sigma = 0$. We point out the qualitative agreement between the phase drift $D''$ deduced from this back-of-the-envelope estimate and our result of Eq. (50). Quantitatively, though, this heuristic description predicts a linewidth which is two times smaller than the real one. There is also a difference of a factor of 4 in the $\sigma$ term. This latter difference is most likely because we considered uncorrelated pairs of correlated atoms (where every other atom is correlated to its follower only), whereas all atoms are correlated to all - in other words, this back-of-the-envelope argumentation assumes no correlation beyond $t > 1/\Gamma$.

Figure 7 summarises the main properties of the continuous superradiant laser following the cumulant approach. In panel (a), the linewidth is shown, for $g/2\pi = 3 \times 10^3$ Hz and $\kappa/2\pi = 1 \times 10^6$ Hz, as a function of the transit rate $\Gamma_R$, for two atom numbers ($N = 500$, black solid line; $N = 20000$, red solid line). For comparison, the results in the mean-field approximation are also shown (dashed curves). For a large $\Gamma_R$, the superradiant emission stops, corresponding to $Ng^2/\kappa\Gamma_R > 1/2$. The unphysical divergence in the mean-field approximation disappears in the cumulant approach. For $Ng^2/\kappa\Gamma_R \gg 1$, $\Delta\omega \approx \Gamma_R$. For $Ng^2/\kappa\Gamma_R \ll 1$, $\Delta\omega \approx 4g^2/\kappa$. Panels (b-c) correspond to $N = 20000$. Panel (b) demonstrates the very large increase of the collective dipole in the superradiant regime. In the cumulant approach where $\langle s^+ \rangle = 0$, such a large dipole arises from atom-atom correlations, and more explicitly, are due to the non-vanishing value of $\sigma$. When $Ng^2/\kappa\Gamma_R \ll 1$, the collective dipole given by $\langle S_x^2 \rangle$ is lowered down to the $N/4$ value corresponding to uncorrelated atoms. Panel (c) shows the value of the photon rate, leaking out of the cavity. For all values of $\Gamma_R$ in the superradiant regime, the rate is very close to $\Gamma/2$, meaning that each atom on average emits half a photon into the cavity mode.

Our results can be compared to previously published results that were obtained for the case of a continuously repumped superradiant laser [16,17], for which a linewidth $\propto g^2/\kappa$ is also obtained. However, we point out that reloading atoms and repumping atoms are not identical. Indeed, repumping atoms removes one atom in the ground state to create an atom in the excited state, while the re-loading approach corresponds to losing one atom (irrespective

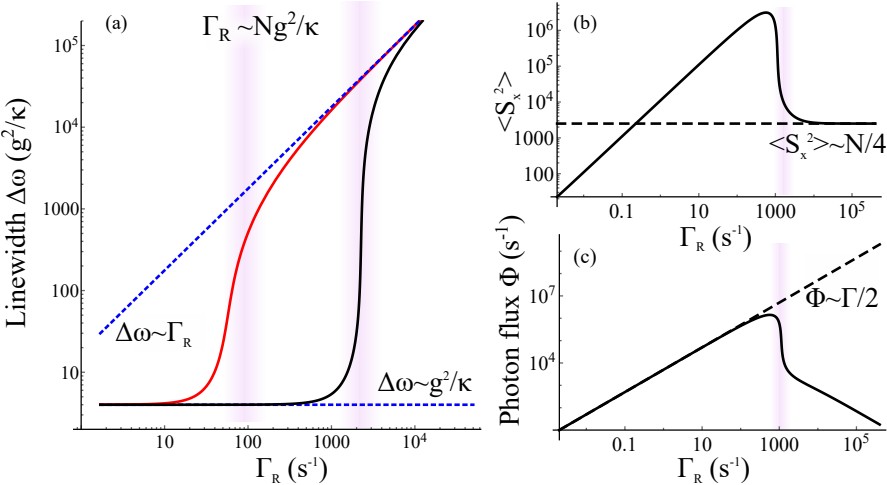

Figure 7: Properties of the superradiant laser. (a) linewidth for two different atom numbers (500, red; $2 \times 10^4$, black), following the cumulant expansion approach. The theory taking into account correlations avoids the pathological divergence of the linewidth that is observed in the classical model at strong $\Gamma_R$. The blue dotted lines show the obtained linewidth in the two limiting cases $N^2 C' \ll 1$ and $N^2 C' \gg 1$. (b) Length of the collective dipole. This length collapses to that of independent random dipoles for large $\Gamma_R$ (represented by the black dashed line). (c) Photon flux exiting the cavity. Deep in the superradiant regime, each atom emits on average half a photon (black dashed line), while for $N^2 C' \ll 1$ the emission of photons is strongly reduced. Shaded areas represent the regime close to threshold where our model may be expected to be inaccurate.

of its internal state) and gaining another atom in the excited state. We also point out that a reduction of the linewidth below the limit set by the Purcell rate $g^2/\kappa$ has also been discussed in [16], potentially related to entanglement between atoms and photons. This is not seen in our model of the beam superradiant laser.

We now compare the width of the superradiant laser $\Delta\nu_{SR} \propto g^2/\kappa$ to the ultimate linewidth of lasers following the Schawlow-Townes limit $\Delta\nu_{ST} \propto \frac{\kappa}{4\pi N_\nu}$, where $N_\nu$ is the number of photons in the lasing mode [30, 31], and to the modified Schawlow-Townes limit for a bad-cavity laser, $\Delta\nu_{ST}^{BC} \propto \frac{\gamma^2}{\pi\kappa N_\nu}$ [27]. To do so, we postulate that each loaded atom emits about one photon in the cavity mode: $N_\nu \propto \Gamma/\kappa$. We find $\frac{\Delta\nu_{SR}}{\Delta\nu_{ST}} \propto \frac{\Gamma g^2}{\kappa^3} \ll \frac{\Gamma_R}{\kappa} \ll 1$, and $\frac{\Delta\nu_{SR}}{\Delta\nu_{ST}^{BC}} \propto \frac{\Gamma g^2}{\kappa\gamma^2} = NC\frac{\Gamma_R}{\gamma} \gg 1$. Therefore, the linewidth of the superradiant laser is smaller than the Schawlow-Townes limit for good cavity lasers, but larger than the usual theory of bad cavity lasers [27]. The fact that the linewidth of the superradiant laser is smaller than the Schawlow-Townes limit is because the narrow-linewidth of lasers usually comes from a large intracavity photon number, and associated reduced phase fluctuations. In our case, however, the ratio of the intracavity photon number to the atom number, $\Gamma_R/2\kappa$, is small and the coherence is established by that of the collective atomic dipole, rather than thanks to stimulated emission. The key point of narrow linewidth for superradiant lasers is indeed the large collective dipole $\sqrt{\langle S_x^2 + S_y^2 \rangle}$ that requests synchronisation between individual dipoles. Finally, the linewidth of the superradiant laser is larger than the modified Schawlow-Townes limit taking into account the bad cavity regime. This corresponds to a third independent regime, discussed in [23], characterised by an atomic transit time shorter than the natural life-time $1/\gamma$.

# 6 Conclusion

We have presented a minimalistic model of an atomic beam CW superradiant laser based on a Hamiltonian approach, in order to describe its salient features. The architecture is a continuous beam of atoms in an excited state crossing the mode of a Fabry-Perot cavity. We show that the individual dipole of an atom entering the cavity tends to align with the collective dipole of the atoms that are already in the cavity, which is the main mechanism that ensures a sustained collective dipole, despite emission of photons and a finite transit time of the atoms. Our approach allows to find analytical solutions that define the conditions for superradiance to occur, and predict the power and linewidth. We find that deep in the superradiant regime, the linewidth is set by the Purcell rate, and is only marginally impacted by atom-atom correlations. Our analytical approach provides a clear physical picture where the ultimate linewidth of the superradiant laser is set by the quantum fluctuations of the atomic collective dipole.

# Acknowledgements

We thank Grégoire Coget, Igor Ferrier-Barbut and Marion Delehaye for stimulating discussions, and their feed-back on our work.

**Funding information** We acknowledge support from Agence Nationale de la Recherche (project CONSULA, ANR-21-CE47-0006-02), Labex FIRST-TF (project SURECO), and Ile de France Region in the framework of DIM SIRTEQ (project FSTOL).

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
