# Peer review of "Correlations and linewidth of the atomic beam continuous superradiant laser"

_SciPost Physics Core, doi:SciPost Phys. Core 6, 015 (2023)_

## Round 1 · Referee Report · Simon Balthasar Jäger (Referee 1) · 2022-10-31

Strengths

  • Nice introduction

  • Comparison of different methods

  • Calculation of various properties of the laser

Weaknesses

  • Key assumption is unclear

Report

The authors Bruno Laburthe-Tolra et al. study in their work ``Correlations and linewidth of the atomic beam continuous superradiant laser'' various properties of a superradiant laser based on an atomic beam. In their manuscript the authors begin by introducing a Hamiltonian description to determine the dynamics and spectrum of the superradiant burst of an initially inverted atomic ensemble coupled to a single-mode cavity. Using and modifying this Hamiltonian description the authors derive the mean-field dynamics of an atomic beam which traverses and couples to an optical cavity. In their calculation they include the constant ``refreshing'' due to new atoms entering the cavity and also free-space spontaneous emission. With this the authors are able to derive the mean-field dynamics and steady state which is used to determine the lasing regime of this device. In addition the authors also use Monte-Carlo simulations, phase diffusion arguments, and second-order cumulant methods to determine the linewidth of the light field. The cumulant methods are also used to quantify correlation within the atomic ensemble.

In my opinion this is a well written draft which is interesting and timely regarding the various groups working on superradiant lasing with both trapped atoms and atomic beam configurations. The introduction is very didactic and I appreciate the use and comparison of different methods to analyze the physical model. Therefore my opinion is that this manuscript is in general well suited to be published in SciPost Physics Core.

However, before giving my final approval I have a few minor comments and also (some) main question(s) that I believe should be addressed and commented on by the authors.

Minor comments:

(a) On page 2 the authors highlight that they are using a Hamiltonian description without stochastic terms. To my eyes this is basically the mean-field approximation. In addition, while I agree that the collective emission can be cast into a Hamiltonian, I would be surprised if Eq.(20) of the manuscript can be cast into a (hermitian) Hamiltonian. Can the authors comment on this?

(b) I believe it would be helpful to add a few definitions such that there is less space for confusion. For instance on page 2 the authors define the spin operator and I think it would helpful to express them in the excited and ground state projectors and also define $s^x$ and $s^y$. Also, I have not found the definition of Var() and CoVar() [page 21].

(c) On page 4 the authors get two terms for the integral above Eq. (5). The second term (real part) results in the delta function in the long-time limit which is used to obtain the cavity damping rate. The first term would usually result in a frequency shift, however, the authors claim that it vanishes because of the summation over $k$ and since the integrand is odd in $\omega_k$. If I understood it correctly, the latter is the detuning between the cavity resonance and the $k$-mode of the free electromagnetic field $c|k|$. I would argue assuming that this integral vanishes exactly depends on the properties of the spectral density. Nevertheless, even if this integral does not vanish it only renormalizes the cavity resonance. Could the authors comment on that and maybe add a sentence how they can drop this term?

(d) On page 5 the authors write ''In other words, the laser spectrum is at best Fourier limited by the pulse envelope, so that only a sustained or CW laser can ever reach the very narrow linewidths that are necessary for metrological applications.'' I agree that the linewidth in the pulsed regime is $Ng^2/\kappa$ and in the CW mode it is of course much narrower. Still $Ng^2/\kappa$ can also be very narrow if the linewidth of the atomic transition is very narrow. Therefore I think the statement that the pulsed mode is not useful for metrological applications appears to me too strong.

(e) On page 7 the authors write ''We do no expect that the actual mathematical form chosen for $\eta_j(t)$ impacts the result of our analysis.'' In the past, we have performed simulations with different mode functions, velocity distributions, and also stochastic atom insertion and they have sometimes changed the quantitative results such as threshold and intensity. However, the qualitative behavior was in our simulations the same. Maybe the authors could add the word ''qualitatively''.

(f) Equation (20) is equivalent to the mean-field equations for superradiant lasing of a trapped atomic ensemble where the atoms undergo spontaneous emission with rate $\gamma$ and are repumped at rate $\Gamma/N$. Therefore all subsequent mean-field results must be the same as the ones describing a trapped-atom superradiant laser. I think this should be highlighted in the text directly after Eq. (20). In fact the authors write later on page 12 about the lasing threshold: ''A similar condition has been found in the case of a continuously repumped atomic sample [4].'' I would argue these are not similar but the same conditions if one believes that Eq. (20) (which describes the mean-field dynamics of a trapped atomic ensemble) accurately describes the dynamics of an atomic beam.

With this last comment I also come to the main question.

Main question(s):

On page 8 the authors write: ''Furthermore, we also make the following key assumption: $\langle s^-_{j-N} \rangle= \langle S^-\rangle /N$ .'' They further argue: ''We expect this to be valid deep in the superradiant regime when the natural timescale for dynamics $Ng^2/\kappa$ greatly exceeds the transit rate $\Gamma_R$ .'' Also in their Monte-Carlo simulations, I believe, the authors seem to use this when they write ''The disappearance of an atom from the cavity is described by a discrete reduction of the average atomic operators: $S^\epsilon\to(N-1)/N\times S^\epsilon$ after dynamics has taken place.'' on page 14. This assumption is also used in the second order cumulants if I am not mistaken.

I would argue that the value of $\langle s^-_{j-N} \rangle $ depends on the history of the particle inside of the cavity. To highlight this let me propose an example: The highest efficiency of this laser is reached if every atom emits exactly one photon into the cavity. This means that the atom makes a Rabi cycle (with phase $\pi$) from its excited state to its ground state (where it has no more coherence) and consequently $\langle s^-_{j-N} \rangle =0$. Still, the mean collective dipole $\langle S^-\rangle\sim N$ is very large since every atom inside of the cavity (away from the boundaries) carries coherence. In a beam configuration I would argue that there is a mean evolution of the internal atomic state along the cavity waist. For a homogeneoulsy driven and coupled trapped atomic ensemble every atom is in average in the same state (basically no spatial dependence on the mean field level). In a beam configuration, instead, there is a spatiotemporal spin configuration which allows superradiant emission although the mean inversion $\langle S^z\rangle$ is close to zero (see example above). I believe, this cannot happen for a superradiant laser in a trapped atomic ensemble because population inversion is crucial there. For me, this is one big difference between superradiant lasers based on trapped atomic ensembles and atomic beams

My question is now: how can you assume that every atom has relaxed to $\langle s^-_{j-N} \rangle= \langle S^-\rangle /N$?
The authors propose two scenarios on page 8: (A) the atoms reach a steady state shortly after entering the cavity. In this case I ask : what should the relaxation process be? I can imagine at least two: spontaneous emission or additional interactions e.g. collisions. In that case relaxation would require that the typical relaxation time is short compared to the transit time. But this seems to me a very bad regime for the laser operation because one could expect that these processes broaden the laser line or even destroy superradiance. (B) The second scenario which is proposed is that the trajectories of the atoms are very different such that due to ergodicity $\langle s^-_{j-N} \rangle= \langle S^-\rangle /N$ holds. In the lasing (superradiant) regime, I would argue exactly the opposite is true. Every atom follows in average the same trajectory (since the initial condition is fixed: all atoms enter in the excited state) and only beyond mean-field effects add fluctuations around this trajectory. During their flight each atom sees in average the same cavity field and makes the same Rabi oscillation according to the field intensity and phase.

I think that it is crucial that the authors comment on and discuss further their ''key assumption''. In fact, I find this very interesting since it allows them to map the beam configuration onto a trapped atom configuration (at least on the mean-field level).

---

## Round 2 · Referee Report · Simon Balthasar Jäger · 2022-11-21

Report
The authors have addressed my comments and questions. Some of their comments are hard to read because of the formatting. Nevertheless I believe that I have understood most of their reasoning. In particular, footnote [24] adds further clarification on the main assumption made by the authors. In my opinion the manuscript can be published as it is.

---

## Round 2 · Author Response

About the referee main question:
First, it is important to clarify that an atom entering the cavity does not merely undergo Rabi oscillations in the cavity field. This picture can be used as a toy model to guess the main physical properties and/or regimes, but does not accurately picture the physics. Rather, the system is a many-body system, and the field is created by the atoms themselves such that the picture of atoms coupled to a pre-existing cavity field should be taken with a grain of salt. This is especially true because in the superradiant regime, the number of photons is smaller than the number of atoms, so that describing atoms as merely undergoing Rabi oscillation in a pre-existent field is necessarily insufficient. As a consequence, it does not appear to us that one can necessarily reach the regime where each atom will send exactly one atom in the cavity.
Let us however stick to this picture to try and answer the referee’s question on why we assume $<s^-_{j-N}>=< S^- >/N$. The Rabi frequency can be written as $\Omega = g b$ where $b$ is the cavity field. Using Eq. 12 and 23, we have $(\Omega/\Gamma_R)^2 \approx N g^2 / \kappa \Gamma_R$. Therefore, in the relevant regime for steady-state superradiance, $\Omega / \Gamma_R >>1$. That is to say: each atom has had the time to undergo many cycles of Rabi oscillation before leaving the cavity after a time $1/\Gamma_R = w_0/v$ (w_0 is the waist of the mode, and v the velocity of atoms). Our main answer to the referee’s question is a follow up on the following sentence from the paper : « the many atoms entering the cavity (in practice at random times and random velocities) follow different trajectories, such that, at the exit, the statistical mean for such random realizations is identical to the average inside the cavity (ergodicity argument). » Let us try and be more precise: if the beam has a spread in velocities $\delta_v$ (as it will always have), the uncertainty on the phase of out-coupled atom is given by $\delta ( \Omega / \Gamma_R) = \Omega / \Gamma_R \times \delta v/v$. Given that $\Omega / \Gamma_R >>1$, a relatively small $\delta v / v$ is sufficient to insure that the atoms leave the cavity at a random phase of their Rabi oscillation, which justifies the claim $<s^-_{j-N}>=< S^- >/N$. We have added a footnote to provide such a reasoning and better justify the approximation (footnote [24]). Note that in addition to this argument, if a 3D cylindrically symmetric cavity mode is assumed (as will be in reality), atoms crossing at different distances from the cavity axis will experience different interaction times, and exit the cavity at different phases of their Rabi cycle.
Note finally that our other guess for a possibility to justify $<s^-_{j-N}>=< S^- >/N$ was « This assumption requires (either) that each atom reaches a steady state shortly after entering the cavity » but did not imply that we had in mind an additional dephasing mechanism compared to those already included in the model. As we explain above, the system cannot be described by a simple Rabi oscillation, and we simply do not know whether the dynamics of a given atom within an ensemble of many atoms can lead to an apparent steady state shortly after entering the cavity. We simply meant to point this as a possibility.
About the referee’s other comments :
(a) Indeed Eq. (20) includes dissipative terms and therefore is not related to a \textit{local} hermitian Hamiltonian. However, our approach is to consider a larger system (of infinite dimension) where, a priori, all atoms that will ever cross the cavity are included, and which also a priori contain all possible electromagnetic modes. Our point is that the system can then formally be considered as Hamiltonian. Only when we focus on the local system at a time t, containing N atoms, do we recover Eq. (20), after some explicit approximations made in the manuscript.
(b) We have now properly defined the spin operators in terms of projectors (now page 3), as well as the variance and covariance (now page 21).
(c) Indeed, the outcome of the integral is in general a bit singular. The key point is that the model only considers propagation in 1D, and in vacuum, so that the density of states is constant and the frequency $\omega_k$ is strictly proportional to $k$. Note however, that things can be more complicated when the coupling $\Omega$ depends on $\omega_k$, which is not taken into account in our model, for simplicity. As the referee himself points out, however, the associated shift only renormalizes the cavity resonances, and therefore does not modify our description of the superradiant laser. In addition to the sentence in our paper “The coherent coupling $\Omega$ of the cavity mode to all these modes is assumed to be independent of k », we now have also written: “The first term is odd in $\omega_k$ so that the summation over $k$ is zero, and the associated frequency shift is therefore neglected.”
(d) We agree that $N g^2/\kappa$ can be small. We have modified the sentence accordingly: ''In other words, the laser spectrum is at best Fourier limited by the pulse envelope. For metrological applications, it can therefore be useful to reach a sustained or CW regime, in order to further reduce the linewidth.''
(e) We have added the word “qualitatively”.
(f) We do not agree with the referee that our equations are identical to the trapped case equations. In the trapped case, repumping only affects ground-state atoms, whereas our loss mechanism (atoms leaving the cavity) is independent of the atomic internal state. This translates into slightly different equations, and different behavior. As expected, the only difference arises from the repumping/loading term. In the trapped case, the repumping leads to $dS^-/dt = -w/2 S^-$ and $dS^z/dt = -w S^-$ whereas in the case of losses $dS^-/dt = -Gamma/N S^-$ and $dS^z/dt = -Gamma/N S^-$. There is a factor of two difference in the ratio of decoherence in the $z$ and $–$ components of the collective spin in the trapped case, which is not the case in our setting. Both cases are therefore different, which is for example highlighted in the paper by the sentence “Indeed, repumping atoms removes one atom in the ground state to create an atom in the excited state, while the re-loading approach corresponds to losing one atom (irrespective of its internal state) and gaining another atom in the excited state.” In addition, on a less formal level, there is also a key practical distinction: repumping $w$ in the trapped case qualitatively corresponds to the loading rate divided by $N$ in the beam case, $\Gamma/N$. Therefore, the atom number scales differently with the tunable experimental knobs; in practice for example, this is why the light intensity scales as $N$ in the beam architecture, and $N^2$ in the trapped case.

---

## Round 2 · List of Changes

In order to answer the referee’s main question, we have added a footnote to better justify the approximation $<s^-_{j-N}>=< S^- >/N$ (footnote [24]).
To address question (b): we have now properly defined the spin operators in terms of projectors (now page 3), as well as the variance and covariance (now page 21).
To address question (c), we have added the following sequence “The first term is odd in $\omega_k$ so that the summation over $k$ is zero, and the associated frequency shift is therefore neglected.”
To address question (d) we have modified a sentence which now reads: ''In other words, the laser spectrum is at best Fourier limited by the pulse envelope. For metrological applications, it can therefore be useful to reach a sustained or CW regime, in order to further reduce the linewidth.''
To address question (e), we have added the word “qualitatively” in the sentence: "We do no expect that the actual mathematical form chosen for \eta_j (t) qualitatively impacts the result of our analysis. "

---

## Editorial Decision

published